# OpenReview forum: "Z0-Inf: Zeroth Order Approximation for Data Influence"
_ICLR.cc/2026/Conference — Submitted to ICLR 2026_

### Official Review · Reviewer_gYoV · 2025-10-31

**Soundness:** 3
**Presentation:** 3
**Contribution:** 3
**Rating:** 6
**Confidence:** 3

**Summary:**

This paper introduces Z0-INF, a zeroth-order approximation for data influence that bypasses expensive gradient computations by using finite difference directional gradients across model checkpoints. This makes it much faster and less memory-intensive while being more accurate for large language models.

**Strengths:**

Broad applicability: As the method does not require differentiable loss functions, it can be applied to black-box or non-standard loss settings, which extends the toolbox for both theory and practice.

Substantial computational efficiency gains: Z0-INF leverages finite-difference (zeroth-order) information to dispense with gradients and Hessian inverses, yielding significant reductions in both memory and runtime.

**Weaknesses:**

Potential missing ablations for checkpoint interval and layer selection. The paper claims computational savings by only considering last-layer parameters; however, it does not report how performance degrades or improves with the inclusion of different subsets of model parameters. Similarly, the trade-off between number of checkpoints and estimation quality is not studied.

**Questions:**

To what extent does the selection of parameters (e.g., only last-layer vs. selected layers or all parameters) affect influence estimation? Is there a noticeable gain in accuracy for higher parameter inclusion, and if so, does it offset the computational gains?

---

> ### Author Response · Authors · 2025-11-21
>
> Excellent Questions!
> We conducted thorough experiments on the checkpoint and neighborhood selection. See the response to the 1st Reviewer’s questions and section 4.3 in the updated paper.
>
> **Layer Selection**
> Overall we do not observe significant changes when we add more parameters.
> We used the last linear layer in our original submission because exact TracIn is not computationally feasible for all training parameters and the last linear layer is a standard alternative proposed by `Pruthi, Liu, et. al. Estimating Training Data Influence by Tracing Gradient Descent`. Figure 4 in our updated paper shows how the correlation accuracy changes insignificantly as we probe first two, self attention, MLP, last linear layer and all model parameters for zeroth-order approximation. Below we also present the results for self-influence in a table for Math and Dolly datasets. It is interesting to observe that layer selection does not significantly impact the correlation accuracy. We know already that the main signal about the influence is carried by the loss given the evidence that the variance of the loss is highly correlated with SSRT for self-influence(Section 3.1).
> The table below shows the correlation between Z0Inf self-Influence for 10 checkpoints for selected layers.
> | Layer         | Dolly Corr (SSRT, Z0Inf) | Math Corr (SSRT, Z0Inf) |
> |---------------|:----------:|:---------:|
> | 1st Self Attn        | 0.912      | 0.822     |
> | 1st MLP       | 0.9119     | 0.8297    |
> | 2nd  Self Attn        | 0.9124     | 0.822     |
> | 2nd MLP       | 0.9119     | 0.822     |
> | Last Linear   | 0.8967     | 0.830     |
> | All           | 0.8993     | 0.830     |

---

### Official Review · Reviewer_K6ic · 2025-11-01

**Soundness:** 3
**Presentation:** 2
**Contribution:** 3
**Rating:** 4
**Confidence:** 4

**Summary:**

This paper introduces Z0-Inf, a fast zeroth-order method to estimate data influence without gradients or Hessians. It achieves higher accuracy and lower cost than baseline methods, demonstrating its effectiveness.

**Strengths:**

1. The paper presents both theoretical analysis and experimental results.

2. The proposed zeroth-order method demonstrates impressive computational efficiency and scalability.

**Weaknesses:**

1. The motivation could be strengthened and validated by providing comparisons with true influence.

2. The self-influence formulation appears very similar to [1].

3. The Spearman correlation results in Section 4.3 are relatively low.

4. The writing can be made more consistent. For example, standardizing equation references ("Equation (4)" vs. "6" in line 427) to improve readability.

5. Figure 3 could be improved by aligning the bars and x-axis more precisely and using clearer visual groupings to better distinguish between datasets.

6. There is a missing citation in Line 113: "Lastly ? proposes…".

```
[1] Chhabra, Anshuman, et al. "Outlier Gradient Analysis: Efficiently Identifying Detrimental Training Samples for Deep Learning Models." ICML (2025).
```

**Questions:**

Please see the Weaknesses part.

---

> ### Author Response · Authors · 2025-11-21
>
> Thank you very much for taking time and reviewing our paper.
>
> 1. > The motivation could be strengthened and validated by providing comparisons with true influence.
>
> We are not sure what the reviewer mean by providing comparisons with true influence.
> Is true influence remove and retrain, influence functions or ground truth human labels ?
> Do you mind, please, elaborating further ?
> If it refers to human labelled ground truth, then human labels for cross-influence is scarce and subjective. There are cases when more than one training example could be equally semantically influential for a test example. It would be very hard to tell which one is more influential. Also, what we humans consider influential might not be considered influential by our models. There are many nuances and the evaluation of influential examples is an open research question.
> We chose remove and retrain as our baseline because most algorithms on influential examples, such as  influence functions, aim to approximate remove and retrain effects.
>
> 2. > The self-influence formulation appears very similar to [1].
>
> Thank you for pointing out to `Chhabra, Anshuman, et al. "Outlier Gradient Analysis: Efficiently Identifying Detrimental Training Samples for Deep Learning Models." ICML (2025).`. We added the citation. There are several main differences between that work and ours.
> a) One of the most important differences is that we do not use exact gradients (no-backward pass) but instead approximate the gradients using first order approximation principles. This makes it feasible for our method to be applicable to any non-differentiable loss. Imagine that the loss function is non- differentiable (e.g. RL fine-tuning that uses LLM judges as graders). So far no existing influence estimation method can be used for non-differential losses except Z0Inf (Our method), especially when it comes to cross-influence estimation.
> b) Our approach is not only effective in identifying detrimental / outlier  training examples with high self-influence but also identifying train-test cross influence.
>
> 3. > The Spearman correlation results in Section 4.3 are relatively low.
>
> Our SubSample and ReTrain (SSRT) experiments show that oftentimes multiple training examples contribute to the test to a similar extent and, hence,often  there is no specific training example that stands out. The previous work also shows that it is not always possible to find one or two training examples that have an outsized impact on a test example. There is, however, a set of train-test pairs where both SSRT and our method are able to identify as most influential. We believe that the correlation with SSRT can increase if we increase the number of trials for remove and retrain, however, fine-tuning LLMs hundreds of times is compute and resource intensive. Section 4.3, does show that the results improve as we move from top10 to top5. Showing that if we narrow down the choice of most influential train-test examples, the correlation between SSRT and our method increases.
>
> Good catches! Thank you! We addressed 4, 5 and 6 in the paper!

---

### Official Review · Reviewer_kFvB · 2025-11-01

**Soundness:** 3
**Presentation:** 2
**Contribution:** 2
**Rating:** 4
**Confidence:** 4

**Summary:**

This paper proposes **Z0-Inf**, a zeroth-order approximation method for estimating data influence, particularly self-influence and train-test influence, in modern machine learning models, including large language models (LLMs). The key insight is that first-order gradient-based approximations (e.g., TracIn) become increasingly inaccurate and computationally expensive as model size grows. Instead, Z0-Inf bypasses gradients entirely by using finite-difference directional gradients derived from checkpoint losses, enabling both faster computation and improved correlation with the “remove-and-retrain” baseline (approximated via Subsample-and-Retrain, SSRT). The authors also introduce a variance-based estimator for self-influence that avoids high-dimensional vector operations. Empirical results on vision models and fully fine-tuned LLaMA-3 8B across reasoning and instruction-tuning datasets show that Z0-Inf achieves higher Spearman correlation with SSRT than TracIn and LoGra, while being significantly more efficient in both time and memory.

**Strengths:**

1. **Practical relevance**: The method directly addresses a critical bottleneck in applying influence functions to LLMs, computational intractability by eliminating the need for gradients or Hessian approximations.
2. **Strong empirical validation on LLMs**: Unlike many prior works limited to small models, the paper evaluates on fully fine-tuned 8B-parameter LLMs, demonstrating clear advantages in both accuracy (correlation with SSRT) and efficiency.
3. **Theoretical grounding**: The zeroth-order formulation is well-motivated by the observed breakdown of first-order approximations in large models (Fig. 1b), and the variance-based self-influence bound is a clever simplification.
4. **Algorithmic optimizations**: Techniques like Gram matrix precomputation and offline loss caching enhance practicality without compromising the core idea.
5. **Reproducibility**: Experimental details (datasets, models, SSRT setup, checkpoint usage) are clearly specified, and code release is promised.

**Weaknesses:**

1. **Limited experimental scale**: While LLM experiments are a strength, the datasets used (GSM8K, MATH-500, Dolly) are relatively small (thousands to tens of thousands of examples). It remains unclear how Z0-Inf scales to *large-scale* fine-tuning scenarios (e.g., millions of examples), which are increasingly common in LLM applications. The reliance on SSRT—which itself becomes infeasible at large data scales—further limits the scope of validation.
2. **Train-test influence results are modest**: The gains for train-test influence are less consistent than for self-influence (e.g., TracIn slightly outperforms Z0-Inf on GSM8K), and the paper acknowledges high noise in this setting. The discussion of why zeroth-order helps here is somewhat superficial.
3. **Checkpoint dependency**: The method still requires storing multiple model checkpoints, which may be prohibitive in extremely large-scale training (though less so than full gradients). The sensitivity to the number and spacing of checkpoints is not thoroughly explored.
4. **Lack of downstream task evaluation**: While correlation with SSRT is a reasonable proxy, it would strengthen the paper to show concrete benefits in applications like data pruning, outlier detection, or targeted fine-tuning (as hinted in the conclusion but not demonstrated).
5. **Limited Novelty**: This work represents only a modest advancement over TracIn. Moreover, zeroth-order methods for influence estimation have already been explored in prior literature, for example,  The Mirrored Influence Hypothesis: Efficient Data Influence Estimation by Harnessing Forward Passes

**Questions:**

See weakness

---

> ### Author Response · Authors · 2025-11-21
>
> Thank you for reviewing our paper and for the detailed feedback. Below we address your concerns. Please, let us know if you have additional questions.
>
> **Limited experimental scale and Lack of downstream task evaluation**
> As we also describe in our paper SubSample and Retrain (SSRT) is not feasible for LLMs on larger datasets. Downstream task evaluations such as data selection could be one possible application to validate the effectiveness of our approach on larger datasets, however, it is separate from this work similar to `Mengzhou Xia, LESS: Selecting Influential Data for Targeted Instruction Tuning`. This is a future work and a separate paper, given different data selection strategies based on the influence scores.
>
> **Train-test influence results are modest**
> Our SubSample and ReTrain (SSRT) experiments show that oftentimes multiple training examples contribute to the test to a similar extent and, hence, often  there is no specific training example that stands out. Thus the ordering of influential examples is not always obvious. The previous work also shows that it is not always possible to find one or two training examples that have an outsized impact on a test example. There is, however, a set of train-test pairs where both SSRT and our method are able to identify as most influential. For example, if there are very similar unique pairs of examples that are both in train and test. If we remove that unique train example then the accuracy on the test drops significantly.
>
> We believe that the correlation with SSRT can increase if we increase the number of trials for remove and retrain, however, fine-tuning LLMs hundreds of times is compute and resource intensive. Section 4.3, does show that the results improve as we move from top10 to top5. Showing that if we narrow down the choice of most influential train-test pairs, the correlation between SSRT and our method increases on MATH-500 and Dolly datasets.
>
> **Checkpoint dependency**
> We added a discussion on checkpoint selection in section 4.3 in our paper and it is also included in the response of the first reviewer’s feedback. Overall we observe that the LLM fine-turing converges very fast and the first 5 to 10 checkpoints capture the main variance in loss and are sufficient to get high correlation accuracy with SSRT.  Figure 7 shows that adding more checkpoints can even slightly increase the noise and reduce the accuracy. This is a general trend that we observe across all fine-tuned LLM models. We believe that during fine-tuning models converge relatively fast since the pre-training already provides strong generic capabilities and fine-tuning makes adjustments with a smaller learning rate.
>
> **Limited Novelty**
> 1. One of the major novelties of this work is that Z0inf is the first influence estimation method (both self- and cross-influence) that can be applied to any loss function (differentiable and non-differentiable). This opens up a wide range of applications for models that have non-differential loss such as RL fine-tuned models where we cannot always compute exact gradients.
> 2. Another important novelty of this work is that we establish a theoretical connection between the variance of the loss and TracIn using zeroth-order approximation of gradients (not exact gradients, no backward pass is required). This connection is not only theoretical but also has strong empirical evidence.
>
> We added a citation to the purposed paper but we would like to argue that `The Mirrored Influence Hypothesis: Efficient Data Influence Estimation by Harnessing Forward Passes` is not a zeroth-order method. It suggests continuously training the model on test samples which requires a backward-pass, exact gradient, first-order gradient computation using the chain rule.

---

### Official Review · Reviewer_Z2t8 · 2025-11-01

**Soundness:** 3
**Presentation:** 3
**Contribution:** 3
**Rating:** 4
**Confidence:** 4

**Summary:**

This paper proposes a **zeroth-order influence estimator (Z0-INF)** that replaces gradient-based stepwise contributions with simple **forward-loss differences between adjacent checkpoints** along the training path. By aligning influence with the **actual update direction** taken during training, the method aims to capture what truly changed the model’s behavior without using backpropagation. The authors evaluate both **self-influence** (per-example impact on the trained model) and **train→test influence** (how a training example affects a target/test example), and provide ablations on checkpoint selection and practical implementation choices. Empirically, Z0-INF delivers strong self-influence results and competitive train→test performance, while being easier to run on large models since it only requires forward passes.

**Strengths:**

- **Clear idea, simple to implement:** Replacing gradients with loss differences along the actual training updates is conceptually neat and engineering-friendly.
- **Optimizer- and path-aware:** By tying scores to the direction the optimizer actually moved, the estimator reflects what happened in training rather than assuming perfect first-order linearization.
- **Forward-only practicality:** No backward/Hessian; easier scaling to large models and long training runs.

**Weaknesses:**

- **Compare Z0-INF with Hessian-free Inner-Product (TracIn) and add a phase view.**
  It would help readers if the paper **explicitly contrasts** Z0-INF with the Hessian-free inner-product family (TracIn-style methods) and gives a **phase-aware** view (early, non-converged vs. late, near-converged).
  *My view (for context only):* Z0-INF is likely stronger **early** when steps are larger and curvature is strong, while Inner-Product (TracIn) may fit better **near convergence**. A related reference that may help frame this comparison is **“Revisit, extend, and enhance hessian-free influence functions”**; citing and contrasting with that line could make the differences clearer.

- **Optimizer / schedule sensitivity.**
  Because Z0-INF follows the **actual update direction**, results can vary with **AdamW vs. SGD**, momentum, weight decay, and learning-rate schedules. Please add controlled ablations. A simple step-level diagnostic (e.g., how well each method tracks loss changes under different optimizers) would make the conclusions more convincing.
  *Expectation:* Under **AdamW-style preconditioning**, Z0-INF may outperform Inner-Product (TracIn) more clearly; with **small-step SGD**, the gap may shrink.

- **Checkpoint density & neighborhood averaging.**
  Influence quality likely depends on **which checkpoints** are kept and **how many neighbors** are averaged. Please include **sensitivity plots** (number/placement of early checkpoints; neighborhood size) and provide **recommended defaults** for practical use.


If these concerns are addressed, I’m willing to raise my score.

**Questions:**

Same with weakness

---

> ### Author Response · Authors · 2025-11-21
> **Phase-wise View**
>
> Thank you for the thoughtful comments and the thorough review. Those are great questions!
> We updated the paper with the requested details (see section 4.3 CHECKPOINT, NEIGHBORHOOD, OPTIMIZER AND PARAMETER SELECTION). Below we present several results in markup tables with additional explanations. Please, let us know if you have additional questions.
>
> **Phase-wise View**
>
> Below we visualize the phase-wise view for TracIn and Z0Inf along with the variance of the loss in each phase for the Dolly dataset fine-tuned on LLaMa 8B model.  We observe that the largest correlation between the SubSample and ReTrain (SSRT) and Z0Inf happens during the first 10 epochs. The variance of the average loss in each phase also shows that trend.
> We observe that the variance of the loss is the largest in the first 10 epochs then it declines significantly.
> It’s also interesting to observe that the correlation between SSRT and Z0Inf is significantly larger compared to the correlation with the TracIn especially in the first epochs.
>
> Our overall recommendation is to compute the zeroth-order for the epochs where the most change in the loss is happening. For the LLM fine-tuning the model converges relatively fast (first 10 epochs) since the model parameters are optimally learned during the pre-training and fine-tuning is making certain adjustments typically with a smaller learning rate  on a specific domain dataset and converges relatively fast.
>
> | Phase | Corr(Z0Inf) | Corr(TracIn) | Avg. Loss Variance |
> |----------|---------------:|----------------:|----------------:|
> | 0-10     | 0.895          | 0.431           | 0.114           |
> | 10-20    | 0.280          | 0.102           | 0.0000456       |
> | 20-30    | 0.259          | 0.076           | 0.000155        |
> | 30-40    | 0.229          | 0.026           | 0.0000407       |
> | 40-50    | 0.164          | -0.035          | 0.0000283       |
> | 50-60    | 0.127          | -0.052          | 0.0000142       |
>
>
> **“Revisit, extend, and enhance hessian-free influence functions” paper.**
>
> It is indeed an interesting work that relies only on the exact gradients of the final checkpoint. To address the challenges of generalizability and non-linear models, it ensembles the scores over multiple models trained using model parameter dropout mechanism.The expectation over multiple local minimas provides a more stable approximation for non-convex optimization landscape.
> Similarly, the training dynamics captures the non-linear approximation of the loss landscape and measures influence along the true (non-linear) optimization path. This is one of the advantages of using multiple checkpoints vs a single checkpoint.
>
> One important contribution of our work is that we do not use the exact gradients (often computed during backpropagation via the chain rule) we solely rely on the first order statistics and approximate the gradients using the Taylor’s expansion theorem. This opens a new field of applications for influence functions where the loss doesn’t need to be differentiable. We cited the paper in the related work section.

---

> > ### Author Response · Authors · 2025-11-21
> > **Optimizer / schedule sensitivity.**
> >
> > **Optimizer / schedule sensitivity.**
> >
> > Since LLM fine-tuning with SGD is time consuming, especially, for SubSample and ReTrain (SSRT), and SGD is not commonly used for LLM fine-tuning we compared SGD vs Adam for CIFAR-10 dataset using a small ConvNet (~707k parameters).
> >
> > Below we present the experimental results in a table ( see updated paper’s Section 4.3).
> >
> > In addition to the correlation scores, we also present loss variances within each checkpoint interval as we increase the number of training epochs. Note that here the results are presented not phase-wise but cumulatively starting from 1st epoch up to the kth epoch.
> >
> > Here `Corr (z0inf) ` refers to the correlation between the SSRT (Subsample and Retrain) and Zeroth-order approximation.  `Corr (TracIn)` refers to the correlation between the SSRT (Subsample and Retrain) and TracIn approximation. We observe that the gap between TracIn and Z0Inf is smaller for SGD and slightly larger for Adam. Overall Adam converges faster and the loss change is more linear for SGD compared to Adam. Overall, we had to increase the learning rate from 0.001 to 0.01 for SGD to be able to obtain comparable model accuracy. Otherwise SGD is converging much slower for the sample learning rate of 0.001.
> >
> > | Checkpoint Intervals | SGD Corr (z0inf) | SGD Corr (TracIn) | SGD Loss Variance | Adam Corr (z0inf) | Adam Corr (TracIn) | Adam Loss Variance |
> > |----------|------------------|-------------------|--------------|-------------------|--------------------|--------------|
> > | 0-10     | 0.571            | 0.558             | 0.586        | 0.737             | 0.705              | 0.741        |
> > | 0-20     | 0.666            | 0.645             | 0.559        | 0.792             | 0.755              | 0.542        |
> > | 0-30     | 0.694            | 0.677             | 0.521        | 0.815             | 0.779              | 0.417        |
> > | 0-40     | 0.705            | 0.690             | 0.487        | 0.824             | 0.788              | 0.317        |
> >
> > It’s also interesting to observe that for LLM fine tuning the loss drops significantly close to zero in the first 10 epochs, while for Covnet the process is more gradual, hence later epochs are more important for the ConvNet use case. We observe this phenomenon for all three fine-tuned LLMs. We included the results for Dolly but can updates the plots with other models as well.

---

> > > ### Author Response · Authors · 2025-11-21
> > > **Neighborhood sensitivity**
> > >
> > > **Neighborhood sensitivity**
> > >
> > > Below we showcase that as we increase the number of neighboring checkpoints the correlation accuracy increases and peaks at 8 neighbors for the dolly dataset.
> > > Here the neighbors |M_i| In Equation 6 are chosen based on the proximity to a given checkpoint. Note that there are different ways of measuring proximity such as L2 distance. We used adjacent indices of the checkpoints to pick the neighbors. Generally, the more neighbors we pick the better but we want those neighbors to be not too far since it might in some cases affect approximation accuracy. The choice of  neighbors can affect the accuracy but we did not observe large changes in the accuracy as we increase the neighbors.
> > >
> > > The experimental results are for Dolly dataset, for the first 10 epochs of the fine-tuning. We will add the numbers for GSM8k and MATH-500 as well.
> > >
> > > | Nearest Neighbors (k) | SpearmanR |
> > > |------------------------|------------|
> > > | 2                      | 0.8599     |
> > > | 4                      | 0.8834     |
> > > | 6                      | 0.8930     |
> > > | 8                      | 0.8967     |
> > > | 10                     | 0.8919     |
> > >
> > > Our overall recommendation is to choose the number of checkpoints based on the change of the loss in the training. Take the checkpoints where the most change in loss is happening. In terms of the neighbors, for LLM fine-tuning 5 - 8 tight neighbors are sufficient but this can depend on optimization landscape and the distances of the checkpoints.

---

> > > > ### Comment · Reviewer_Z2t8 · 2025-11-27
> > > >
> > > > The authors’ response has addressed my concerns, and I have raised my score.

---

### Meta-Review · Area_Chair_N2eA · 2025-12-24

**Summary:**

This paper presented a zeroth-order influence estimator by computing forward-loss differences between adjacent checkpoints, which are the first data influence method that are applicable to all differentiable and non-differentiable models. This study also demonstrated strong empirical validation results on LLMs.

However, the reviewers raise several core concerns including (1) Low practical correlation, (2) Limited empirical scope, and (3) Modest advancement. The author’s rebuttal have addressed part of the concerns, while outstanding issues in presentation and lack of integrity are considered critical for accepted at the conference.

The reviewers raise several core concerns including (1) Novelty and distinctiveness compared with existing works, (2) Marginal and inconsistent performance gains in train-test influence estimation, (3) Dependence on experiment configurations, specifically on checkpoint setting, selection of optimizer and training phases, and （4） Limited experimental scale and evaluation scope on downstream task and alternative influence proxies.

The authors’ rebuttal attempts to address the reviewers’ concerns and some of them have been answered clearly. However, several outstanding issue remain: (1) The evaluation relies exclusively on correlation with SSRT as a proxy for true influence without complementary task-level or behavior-level validations. (2) The relatively modest train-test influence Spearman correlations and burden checkpoint storage suggest the constraints of its practical utility and future applications. (3) One core motivation of the paper is about scaling the data influence estimation to LLMs setting but the empirical evaluation remains limited in dataset scale.

In summary, while the paper presents a conceptually simple and practically motivated approach to zeroth-order influence estimation, the remaining issues still unaddressed through authors’ rebuttal. As a result, I do not find that the current version of the paper meets the acceptance bar.

**Reviewer Concerns:**

I think the authors have addressed part of the concerns, while some major concerns are still outstanding.

In the rebuttal, the authors have partially addressed some of the reviewer concerns, including:

- Novelty and distinctiveness: The authors have made clear justification on the key differences from existing works, showcasing the computation efficiency and theoretical bases of Z0-Inf.

- Improved ablation experiments: The authors have conducted extensive ablation experiments on the effect of checkpoint numbers and intervals, training phase, optimizer types, and layer selection, providing valuable evidence on the method effectiveness and useful practical advice.

- Optimizer/Checkpoint Dependence: The authors have supplied additional results on the method’s sensitivity to optimizer choices, different schedules, number of checkpoints and layers to resolve Reviewer Z2t8 and gYoV’s concern.

- Presentation: The authors have revised the manuscript with advanced formatting and interpretation to address the reviewers’ concern on writings.

- Phase-wise View: The authors have provide the additional experiment results on the phase-wise view for Trac-In and Z0-Inf to understand the choice on timing of computing data influence to address Reviewer Z2t8’s question.

- Related Works: The authors have discussed and compared their study to several related works outlined by the reviewers during the rebuttal stage.

However, there are still some concerns that are still outstanding:

- Incremental Work: Reviewer kFvB have raised the concern that this work represents a `modest advancement over TracIn’. Reviewer K6ic also pointed out that the self-influence formulation appears very similar to one related work. The authors re-stated the contribution lies in the first zeroth-order influence method and the presented theoretical connection. The concerns on work novelty remains outstanding during the rebuttal.

- Limited Empirical Scale: While the paper motivates Z0-Inf as a scalable alternative for LLMs, the empirical evaluation remains confined to relatively small-scale datasets. This gap between the stated motivation and the experimental validation weakens the evidence of proposed scalability.

- The modest validation result: The authors have claimed that the train-test influence results are not always obvious and believe additional remove-and-retrain trials can increase the score. However, such trials are infeasible for LLMs due to the high computational cost, which limits the practical utility and potential applications of the method.

**Reviewer Scores:**

Reviewer Z2t8: 4 -> 6

- I think the authors have partially addressed the raised main concerns by providing the additional results during the rebuttal stage. However, the supplied results on neighborhood sensitivity does not perfectly resolve the question on “which checkpoints are kept” and the the influence of early/late checkpoints.

Reviewer kFvB: 4 -> 4

- I think the authors have addressed the reviewer’s concern on W2-4. Nevertheless, the authors’ response does not extend the experimental scale or answer how this study is not only a modest advancement over TracIn. I believe these two concerns are not fully resolved during the rebuttal stage.

Reviewer K6ic: 4 -> 4

- I think the authors have addressed the reviewer’s concern on W4-6. Nevertheless, all questions regarding W1-3 are not fully explored or discussed during the rebuttal stage, and the outstanding concerns leads to low confidence in acceptance.

Reviewer gYoV: 6 -> 6

- I think the authors have addressed the reviewer’s concerns on missing ablations and question on layer selection.

---

### Decision · Program_Chairs · 2026-01-26

Reject